# Riboswitches as Drug Targets for Antibiotics

**DOI:** 10.3390/antibiotics10010045

**Published:** 2021-01-05

**Authors:** Vipul Panchal, Ruth Brenk

**Affiliations:** Department of Biomedicine, University of Bergen, Jonas Lies vei 91, 5020 Bergen, Norway

**Keywords:** antibacterial drug target, riboswitch, structure-based drug design, fragment screening, high-throughput screening

## Abstract

Riboswitches reside in the untranslated region of RNA and regulate genes involved in the biosynthesis of essential metabolites through binding of small molecules. Since their discovery at the beginning of this century, riboswitches have been regarded as potential antibacterial targets. Using fragment screening, high-throughput screening and rational ligand design guided by X-ray crystallography, lead compounds against various riboswitches have been identified. Here, we review the current status and suitability of the thiamine pyrophosphate (TPP), flavin mononucleotide (FMN), *glmS*, guanine, and other riboswitches as antibacterial targets and discuss them in a biological context. Further, we highlight challenges in riboswitch drug discovery and emphasis the need to develop riboswitch specific high-throughput screening methods.

## 1. Introduction

The emergence of bacterial resistance to antibiotics has become an urgent and serious threat to global public health. Resistant bacteria are expected to cause nearly 10 million deaths each year globally by 2050 [1]. As many pathogenic bacteria are evolving to persist against all existing antibiotics, it is feared that the health system will become incapacitated against serious bacterial infections [2,3]. Indeed, to address this concern, the World Health Organization (WHO) has released a priority list of bacterial pathogens that need immediate attention, to focus antibiotic drug discovery efforts [4]. Consequently, a need for a new generation of antibiotics with novel mechanisms of action against resistant bacteria is increasingly being recognized [5] and current drug discovery programs are exploring both proteins, as well as nucleic acid targets [6,7].

Riboswitches are a novel antibacterial drug target class that could deliver urgently needed antibiotics via a new mechanism of action. They occur almost exclusively in bacteria and regulate the biosynthesis and transport of amino acids and essential metabolites, such as coenzymes, nucleobases and their derivatives by binding small molecules [8,9]. Residing in the 5′ untranslated region (UTR) of mRNA, these *cis*-regulatory elements are structured non-coding RNAs that adopt alternative 3D-conformations to function as genetic switches (“ribo-switching”) [10,11,12]. Riboswitches are comprised of two domains; an aptamer domain that selectively binds the cognate ligands and an expression platform that translates the presence of ligands into expression or repression of downstream genes (Figure 1). When the concentration of a cognate ligand in the intracellular milieu increases beyond a threshold, its binding to the aptamer domain induces a conformational change. In most cases, this ligand-induced alternate conformation leads to transcriptional attenuation by the formation of a terminator or inhibition of translation by sequestering the ribosomal binding site (RBS) or both, followed by switching off the downstream gene or operon [13,14,15]. These riboswitch elements are termed “OFF-switches” (Figure 1a). In other cases, the ligand-induced alternate conformation leads to the formation of an anti-terminator stem and/or release of the RBS from the terminator stem followed by activation of the downstream target gene(s). Accordingly, these are referred to as “ON-switches” (Figure 1b). 

Riboswitches are widespread in bacteria. Currently, 28 experimentally validated riboswitch classes are known, which are distributed across more than 6000 bacterial species [16,17]. Riboswitches are categorized based on their cognate ligand and the fold of their aptamer domain (Table 1). Biochemically, these natural ligands span a wide spectrum of biomolecules, namely coenzymes, nucleotides and their derivatives, ions, amino acids, phosphorylated sugars and guanidine. Some riboswitches are widely present across human bacterial pathogens and therefore, are suitable targets for broad-spectrum antibiotics, while less widespread riboswitches are potential targets for selective antibacterial drugs. The thiamine pyrophosphate (TPP) riboswitch is the most prevalent riboswitch and is the only one present in eukaryotes, including algae [18], fungi [19,20,21] and plants [22,23]. Other riboswitches prevalently present across human bacterial pathogens (>20) include the flavin mononucleotide (FMN), cobalamin, glutamine-fructose-6-phosphate (*glmS*), lysine, S-adenosyl methionine-I (SAM-I) and glycine riboswitches. 

Riboswitches bind to their cognate ligand mostly with high affinity (K_D_ < 10 nM) and discriminate strongly against related compounds (drop in affinity by around two orders of magnitude) [25,26,27]. This suggests the possibility to rationally design potent structural or functional analogues of the natural ligands that can be exploited to starve bacterial cells of essential metabolites and thereby can function as novel antibacterial drugs. To harvest the full potential of riboswitches as antibacterial targets, detailed structural and biochemical knowledge is crucial. The sustained interest in the characterization of various riboswitches has led to a gradual improvement in riboswitch structure wealth over time. Currently, more than 300 crystal structures of riboswitches are available in the Protein Databank (PDB) and at least one representative of most riboswitch classes has been characterized. This has revealed an architectural diversity among different riboswitch classes, leading to a broad categorization into two groups: pseudoknotted and junctional riboswitches [28,29,30,31]. Structural principles of selective binding to distinct riboswitches are now well understood which has in some cases contributed to the rational design of novel ligands [32,33,34].

In order to discriminate cognate ligands from related compounds in vivo, riboswitches have evolved to form a binding pocket that specifically recognizes their natural ligands. In fact, different riboswitches employ distinct strategies to achieve a high affinity and an appreciable degree of selectivity. Some riboswitches, like the lysine and purine riboswitches, possess binding pockets with perfect shape complementarity to cognate ligands and encapsulate them almost entirely and thus exclude binding of related compounds [35,36]. Often, hydrogen bonding with the ligand is an important means of achieving high specificity. This is exemplified by changing the specificity of the adenine riboswitch from adenine to guanine by mutating U74 in the binding pocket to C74 and thus requiring a different ligand to form a Watson–Crick base pair [37]. In addition to nucleobases, also the ribose and phosphate groups of the nucleotides are involved in recognizing the ligands, albeit this varies depending on the riboswitch. The interactions with the negatively charged group of ligands (e.g., a phosphate group in the case of TPP and FMN, Table 1) are often mediated by cations, such as divalent Mg^+2^ and monovalent K^+^, either directly through coordinated bonds or indirectly through water molecule(s) [38,39]. Positively charged groups of ligands, such as those in lysine or SAM, typically interact directly with the negative phosphate backbone and other surrounding polar but neutral atoms of the base or sugar moieties [35,40]. Pi–pi interactions between aromatic rings of the ligands and nucleobases are another driving factor for ligand binding. Altogether, the diversity of interaction patterns together with the shape diversity of the binding pockets allow the different riboswitches to bind diverse ligands with high specificity.

From a drug discovery point of view, not all riboswitches are of equal interest. First of all, in order to be pharmacologically relevant, the riboswitch needs to be present in pathogenic bacteria, control the expression of the essential gene(s) and act as an off-switch. Riboswitches that occur in a range of pathogenetic bacteria offer the opportunity to develop broad-spectrum antibiotics while riboswitches that are not widely spread can be suitable targets for narrow-spectrum antibiotics that are sought after to avoid detrimental effects on the microbiome and to lower the risk of spreading resistance [41]. Second, the binding site of the riboswitch needs to have a pocket that is suitable to bind a drug-like ligand with high affinity. While some binding sites are too small, too polar or too solvent exposed to be suited, others have properties that render them suitable candidates for drug discovery efforts [42,43,44].

Here, we review the drug discovery efforts against various promising riboswitches and discuss their current status as suitable antibacterial drug targets. Further, we highlight challenges in riboswitch drug discovery and emphasis the need to develop riboswitch specific high-throughput screening methods.

## 2. TPP Riboswitch Ligands

The TPP riboswitch is the most abundant riboswitch. This riboswitch regulates the expression of genes involved in the synthesis and transport of TPP (Figure 2a) [16,17,45,46] which is an essential cofactor for the carboxylation and decarboxylation of various metabolic intermediates in the carbohydrate and amino-acid metabolism [47]. The TPP riboswitch is probably the most diverse riboswitch with respect to the mechanisms for regulating gene expression. These include modulating mRNA decay, Rho dependent transcription termination, alternate splicing, transcription termination through the formation of an intrinsic terminator and translation inhibition through sequestration of ribosomal binding sites (RBS) [14,48]. This off-switch is present in 48 human pathogens [49], including most of the pathogens of the WHO priority list (Table 1). Thus, this riboswitch is one of the most suitable candidates for broad spectrum antibiotics.

A compound that established the proof of principle for the TPP riboswitch as an antibacterial drug target became available well before the riboswitch was discovered. Pyrithiamine (PT) (Figure 2a), originally designed and synthesized as a structural analogue of thiamine to study thiamine metabolism, is toxic to fungi as well as bacteria [45,50,51]. PT becomes phosphorylated intracellularly to form pyrithiamine pyrophosphate (PTPP) (Figure 2a). PTPP has a similar binding affinity to the *B. subtilis* TPP riboswitch as TPP in vitro (160 nM vs. 50 nM) and turns off the expression of downstream gene(s) involved in TPP biosynthesis or transport. Further, bacteria resistant to PT were found to have mutations predominantly in the genetic region coding for the TPP riboswitch, hinting that the TPP riboswitch is the target for this compound.

Meanwhile, crystal structures of the TPP riboswitch aptamer domain in complex with TPP as well as other ligands were reported at a resolution ranging from 2.05 Å to 3.2 Å [26,32,38,54]. These structures have played a crucial role in revealing the binding modes of the ligands and for suggesting strategies to explore structure–activity relationships (SAR). TPP binds to the riboswitch in an extended conformation (Figure 2b). The aminopyrimidine moiety of TPP is sandwiched between the bases of G42 and A43 and forms hydrogen bonds with polar entities of G40 and G19. The pyrophosphate moiety of TPP is coordinated to two Mg^+2^ ions and also forms direct hydrogen bonds with C77 and G78. The hexa-coordinated Mg^+2^ ions in this pocket are positioned by direct and water-mediated hydrogen bonds with the surrounding residues and thereby facilitate the binding of the negatively charged phosphate group. The thiazole group appears to be less constrained and only forms hydrophobic contacts with G72. The nucleotides in close contact with TPP are highly conserved across TPP riboswitch sequences, suggesting that potential ligands will bind to all TPP riboswitch binding sites.

Initial efforts to identify novel TPP riboswitch binders were made using a fragment-based approach. Using [^3^H]thiamine dependent equilibrium dialysis, Cressina et al. screened 1300 fragments for their ability to displace a radioligand [55]. A total of 20 hits were identified in the initial screen. However, further validation using a combination of NMR spectroscopy, isothermal titration calorimetry (ITC) measurements and counter-screening against the lysine responsive (lysC) aptamer, the hits were reduced to 17 structurally diverse compounds with dissociation constants in the range of 22–670 μM. None of these fragments were able to reduce the expression of a luciferase gene used as a reporter in an in vitro transcription termination assay (IVTT). The failure to suppress the expression of a downstream reporter gene upon small molecule binding could either be caused by the fragments not being able to induce the conformational change essential for switching off the expression or by the fragments not binding strong enough to the longer transcript used in the IVTT assay.

As a first step towards the rational elaboration of the fragment hits, Warner et al. determined crystal structure of four fragment hits (2QB, 2QC, SVN, and HPA, Figure 2a) bound to the TPP riboswitch [32]. Occupying the pocket within the aminopyrimidine sensor helix of the TPP aptamer, each of these fragments was found to induce a conformational change in G72, while keeping the overall conformation of the aptamer similar to that of the TPP bound form (Figure 2c). The determined structures are valuable tools to facilitate structure-guided fragment elaboration to increase their binding affinities.

Alternatively, structural analogues of TPP have also been developed. By systematically replacing the two rings and the pyrophosphate tail of TPP with analogues, Chen et al. established the SAR for these compounds (Figure 2a) [52]. The replacement of the aminopyrimidine ring with various heterocyclic rings significantly compromised the binding affinity. However, the replacement of the thiazolium ring with a related 5-member heterocyclic ring only modestly affected the affinity. In fact, positively charged heterocycles are well tolerated in this position whereas, replacing the thiazole ring of TPP with open-chain analogues severely compromised the binding affinity, suggesting that a ring structure in this position is important. As expected, the negative charge of the pyrophosphate tail is vital to attain high affinity as replacing the tri-anionic group with bi- or mono- anionic groups reduced the binding affinity by at least two and four orders of magnitude, respectively. Additionally, the pK_a_ of the tri-anionic group is also crucial as the replacement of the pyrophosphate group (pK_a_ ~5.77) with a methylenediphosphonate (pKa ~7.45) improved the affinity by a magnitude of two orders. Encouragingly, ligand affinity and the riboswitch-mediated repression of the reporter gene appeared positively correlated. TPP analogues with the affinity comparable to that of TPP repressed the gene expression to the same extent as TPP, whereas those with mediocre affinity repressed the reporter gene to an extent similar to the weaker ligands. Lünse et al. further investigated the binding of the TTP riboswitch to triazolethiamine (TT) (Figure 2a) and a series of its derivatives using an intracellular gene reporter assay [53]. TT was still the most potent compound with an IC_50_ of 91 μM whereas, that of its derivatives range from 20 µM to the high micromolar. In contrast to the TT, whose potency is dependent on active transport and an in vivo phosphorylation, its methanesulfonate analogue (TTMS) is interesting despite with weaker IC_50_ value as it appears to function independently of intracellular enzymes (Figure 2a).

Altogether, these studies demonstrate that only a few modifications of TPP are allowed to maintain high affinity. Further, phospho-mimicking compounds could be an alternative approach to develop TPP riboswitch dependent effective antibacterial agents. Finally, the discovered fragment hits offer opportunities to develop ligands that are dissimilar to TPP.

## 3. FMN Riboswitch Ligands

The FMN riboswitch regulates the expression of genes involved in biosynthesis and transport of riboflavin (vitamin B12, Figure 3a) [56,57]. As a precursor of the coenzymes FMN (Figure 3a and flavin adenine dinucleotide (FAD), riboflavin plays an essential role in oxidative phosphorylation, β-oxidation and the Kreb’s cycle [58,59,60]. Although riboflavin and FAD also bind to this OFF riboswitch, only FMN associates with high enough affinity to repress the expression of the downstream gene(s) either by transcriptional termination or by translation inhibition. The FMN riboswitch is the third most widespread riboswitch known to date and is found in 41 human pathogens, including seven from the WHO priority pathogen list (Table 1) [13,17,49]. In brief, its essential role in regulating bacterial physiology, the high conservation of the ligand binding domain, the absence of a counterpart in humans and its high abundance make the FMN riboswitch a promising target for broad-spectrum antibacterials.

The FMN riboswitch is an established target for the antibacterial compound roseoflavin (Figure 3a). Roseoflavin (Figure 3a) is a natural riboflavin analogue synthesized by *Streptomyces davawensis* [65,66]. In its phosphorylated form (RoFMN), roseoflavin inhibits the growth of *Bacillus subtilis*, *Escherichia coli*, and *Listeria monocytogenes* by targeting the FMN riboswitch [65,67,68]. This leads to the suppression of riboflavin biosynthesis and transport, and thus the cells are starved of riboflavin. RoFMN binds to the FMN aptamer domain with a similar affinity as FMN (< 5 nM), resulting in the repression of the regulated genes [69,70]. As RoFMN is structurally highly related to FMN, it also binds to flavoenzymes. Therefore, the antibacterial effect of RoFMN is mediated by both the FMN riboswitch and flavoenzymes. Indeed, in roseoflavin-resistant mutants, mutations are found in the FMN riboswitch aptamer domain and flavoenzymes [39,62]. Despite RoFMN not being selective, its FMN riboswitch-mediated antibiotic effect underscores the promise of this riboswitch for antibacterial drug discovery.

The crystal structures of the FMN riboswitch in complex with FMN analogues have been determined with resolutions of around 2.9 to 3.45 Å. These structures revealed that the isoalloxazine rings of the ligands form similar interactions with the aptamer domain. In the complex with roseoflavin, U61 only modestly reorients in order to accommodate the relatively bulkier dimethylamino group in of ligand (Figure 3b) [39]. In contrast, differently substituted alkyl chains in the R^1^ position adopt different conformations in the various structures (Figure 3c). In fact, the ribityl moiety was found to have minimal contribution to the affinity and is discussed in detail in the following paragraph. In the case of FMN, the terminal phosphate group interacts with the aptamer domain through hydrogen bonds directly as well as indirectly through Mg^+2^ ion. These interactions appear to be crucial for binding as riboflavin and roseoflavin, which lack the phosphate group, have a 1000 and 100-fold lower affinity, respectively [39,62,63]. In contrast to the TPP riboswitch, which undergoes a considerable conformational change during ligand binding, various structural and chemical probing methods show that in the FMN riboswitch, the ligand binding pocket adopts a similar conformation in both the absence and presence of FMN [39,63,71,72]. Altogether, the ligand binding region of the FMN riboswitch offers a balance of hydrophobic and hydrophilic interactions and has been predicted to bind drug-like ligands using computational methods [42,43,44].

Recently, a few promising small molecules with antibiotic activity have been discovered to selectively target the FMN riboswitch using three different approaches: a structure-based approach, a phenotypic screen, and high-throughput screening (HTS).

Vicens et al. employed a structure-based approach to identify RoFMN analogues that would not require intracellular phosphorylation for antibacterial activity [34]. Simplifying the ribityl group of RoFMN to an alkyl chain but maintaining the negative charge at its end led to the compounds BRX830 (IC_50_ ~2.6 nM) and BRX1151 (IC_50_ ~8 nM) with similar inhibitory activity as RoFMN (IC_50_ ~3 nM) in an in vitro transcription assay (Figure 3a). A further modification to contain an aromatic side chain led to the compounds BRX1354 (IC_50_ ~16 nM) and BRX1555 (IC_50_ ~39 nM), and eventually the drug-like molecule 5FDQD (Figure 3a). The latter no longer requires a negative charge for potent inhibition of the transcription reaction (IC_50_ ~7.5 nM). Further, analysis through X-ray crystallography showed that the compounds BRX1151, BRX1354 and BRX1555 bind to the aptamer domain in a similar fashion as FMN, with their isoalloxazine ring intercalated between A48 and the A85-G98 pair and hydrogen bonding with A99 (Figure 3c). Interestingly, the binding pocket where the FMN ribityl side chain is located is rather promiscuous as a variety of functional groups on the alkyl chain at R^1^ could be accommodated between G10, G11 and G62 (Figure 3c). Further, the discovery of these compounds clearly demonstrates that the negatively charged chain of FMN could be replaced with hydrophobic moieties without compromising affinity, resulting in potent drug-like ligands. An in vitro characterization of 5FDQD revealed potent and rapid bactericidal activity against *Clostridium difficile* with only a modest effect on the culturable cecal flora of mice [73]. In a mouse model of *C. difficile* infection, the antibacterial performance of 5FDQD is similar to that of fidaxomicin and vancomycin until 8 days post-infection. Despite the promising results of 5FDQD, it is noteworthy that its definitive mode of action remains elusive as no resistant mutants could be obtained to confirm the target.

Another example of a potent FMN riboswitch ligand is ribocil (K_D_ = 13 nM, Figure 3A), discovered serendipitously through a phenotypic screen [61]. The screening involved testing a library of ~57,000 synthetic compounds against an *E. coli* strain defective in lipopolysaccharide synthesis and drug efflux (MB5746). The only hit selective to riboflavin synthesis in the screen was ribocil. Characterization of the ribocil resistant mutants mapped all base pair changes to the FMN riboswitch upstream of the *ribB* gene (involved in riboflavin biosynthesis), establishing it as the sole target. Ribocil is a racemic mixture of the R- (ribocil-A) and S-enantiomer (ribocil-B), and it is the latter that predominantly inhibits the riboflavin biosynthesis and consequently bacterial growth. A further structural elaboration led to the compound ribocil-C (Figure 3a), which has approximately eight-fold higher potency against the Gram-positive bacteria *E. faecalis* and methicillin-resistant *S. aureus* [74,75]. Interestingly, in a murine *E. coli* (MB5746) septicaemia model of infection, ribocil-C demonstrated a dose-dependent reduction in the bacterial burden without any evident toxic side effects. Recently, Motika et al. modified ribocil-C, which lacks activity against Gram-negative pathogens, to ribocil C-PA (Figure 3a), which is also active against Gram-negative bacteria [64]. In fact, ribocil C-PA inhibited all multi-drug resistant clinical strains of *E. coli* and *K. pneumonia* tested in the study. Ribocil C-PA is 16-fold more potent against strains of *E. coli* (MIC = 4 μg/mL), *Enterobacter cloacae* (MIC) = 4 μg/mL), and *K. pneumoniae* (MIC = 4 μg/mL) than ribocil C (minimum inhibitory concentration MIC > 64 μg/mL). Encouragingly, ribocil C-PA was also effective in mice in *E. coli* infection models for acute pneumonia and sepsis. Indeed, ribocil_C-PA rescued 80% of mice infected with pathogenic strains of *E. coli*, whereas ribocil-C was ineffective. Unfortunately, both ribocil C and ribocil C-PA led to a high frequency of resistance (FOR) in *E. coli* strains (at the order of 10^−6^).

More recently, Rizvi et al. demonstrated the application of the affinity selection mass spectrometry (AS-MS) based automated ligand detection system (ALIS) for the discovery of selective riboswitch ligands. They used a HTS approach to screen a library of ~53,000 unbiased small molecules against the FMN riboswitch [76]. Although the hit compounds WG-1 (K_D_ not determined, Figure 3a) and WG-3 (K_D_ ~130 nM, Figure 3a) bind to the FMN riboswitch in an FMN competitive manner, both exhibited riboflavin independent antibacterial activity against *E. coli* (MB5746). This suggests that both are non-selective in terms of their mode of action, and therefore lack potential as FMN riboswitch targeting antibacterials.

Altogether, these studies demonstrate the potential of the FMN riboswitch as a promising target for the next generation of antibiotics, however, high FOR associated with the FMN riboswitch ligands remains a serious concern.

## 4. *glmS* Riboswitch Ligands

The *glmS* riboswitch is unique as it is a self-cleaving ribozyme that uses the cognate ligand—glucosamine-6-phosphate (GlcN6P, Figure 4a)—as a coenzyme to catalyse the cleavage reaction. The riboswitch is predominantly present in Gram-positive bacteria (Table 1) and resides in the 5′-UTR of the glutamine-fructose-6-phosphate aminotransferase (*glmS*) gene, which catalyses the synthesis of GlcN6P [77,78]. At micromolar concentration, GlcN6P induces *glmS* riboswitch self-cleavage to produce an upstream fragment with a 2′,3′-cyclic phosphate (2′,3′-cP) end and a downstream fragment with a 5′-OH moiety (Figure 4b) [79,80]. The latter is rapidly degraded by RNase J1 in *B. subtilis*, suggesting that the *glmS* riboswitch affects the mRNA stability in a GlcN6P dependent manner [81]. Notably, the catalytic core of the riboswitch is conserved across bacterial species (>97% identity), which allows it to function in the same way despite sequence and structure variability outside of the binding pocket. As GlcN6P is an essential metabolite for normal growth [82,83] and an essential precursor for the synthesis of the bacterial cell wall [84], the *glmS* riboswitch is regarded as a promising target to develop antibacterial drugs against various bacterial pathogens (Table 1).

McCarthy et al. were the first to characterize the ligand requirements to induce *glmS* riboswitch self-cleavage using a series of amine-containing analogues. They revealed that the amine functionality of GlcN6P is essential for catalysis [85]. This observation was also corroborated by various structural studies [86,87,88,89]. Effectively, there is no conformational or positional rearrangement of the active site upon binding the ribozyme inhibitor glucose-6-phosphate (Glc6P) or GlcN6P (Figure 4c). Further, comparison of the apo form with the Glc6P or GlcN6P bound *glmS* riboswitch structures showed that the ligands bind to a preorganized active site wherein the nucleophile for the catalysis (2′-OH group of A(-1)) and the leaving group (5′-phosphate group of G1) are in close proximity (Figure 4c). Consequently, binding of GlcN6P activates the riboswitch through the positioning of its primary amine close to the 5′-O of G1 to function as a general acid accelerating the cleavage reaction. It is important to note that replacing the phosphate group of Glc6NP with a sulphate group reduced the affinity by 100-fold, indicating that the phosphate group is critical for binding.

Up to now, efforts to explore the *glmS* riboswitch as an antibacterial drug target are focused on hit discovery and SAR studies. Evaluation of the discovered compounds against pathogenic bacteria is still outstanding. The Breaker group developed a fluorescence resonance energy transfer (FRET) based high-throughput assay to screen structural and stereochemical analogues of GlcN6P and a library of 960 compounds approved for use in humans [90,91]. Although no hit was identified from the latter, screening of Glc6NP analogues revealed the importance of various functional groups to induce *glmS* ribozyme activity summarized below (Figure 4a). Using a fluorescence polarization-based assay Mayer et al. screened a library of > 5000 compounds which follow Lipinski’s rule of five and novel analogues of GlcN6P. The latter included various 1-deoxy, 1-methyl glycoside, and carba-derivatives of Glc6NP. None of the compounds from the diverse library and none of the 1-deoxy and 1-methyl glycoside derivatives, which also lacked one or more equatorial hydroxyl moieties were active. In contrast, the carba-analogue of Glc6NP efficiently induced the self-cleavage of the *glmS* riboswitch, suggesting that the ring oxygen is dispensable for the ribozyme activity (Figure 4a) [92,93]. More recently, the group explored mono-fluorinated carba variant of GlcN6P [94]. The most effective compound from the study- fluoro-carba-a-D-GlcN6P (Figure 4a), had a significantly lower activity compared to its non-fluorinated analogue (EC_50_ = 300 μM vs. 6.2 μM), rendering this compound series less promising. In addition, a series of phosphate analogues of GlcN6P were also designed and analysed for their ability to induce *glmS* ribozyme activity [95]. Among these, 6-deoxy-6-phosphonomethyl and 6-O-malonyl ether analogues (Figure 4a) demonstrated potent ribozyme activity.

Taking all studies together, a comprehensive SAR of *glmS* riboswitch activation has emerged. The ring oxygen is non-essential, whereas OH groups are important for the activity as they interact with RNA through hydrogen binding (Figure 4b). Although, due to inconsistent data, it is difficult to compare the affinity of potent analogues discovered through different studies, carba-GlcN6P and 6-deoxy-6-phosphonomethyl and 6-O-malonyl-ether analogues of GlcN6P appear to be the most promising compounds.

Notably, none of these compounds are yet tested for their antibacterial effect or target selectivity. All discovered *glmS* riboswitch activators are GlcN6P-analogs, and screening of diverse libraries did not reveal any hits. However, these libraries were rather small. Perhaps, more hits could be found by screening larger and more diverse compound collections. Conclusively, the feasibility of developing antibacterial agents by targeting *glmS* riboswitch can largely be considered under-explored.

## 5. Guanine Riboswitch Ligands

The guanine riboswitch resides in the 5′-UTR of genes involved in the transport and biosynthesis of purine nucleotides [25]. This riboswitch is moderately widespread and present in two of the WHO priority pathogens; *Staphylococcus aureus* and *Streptococcus pneumoniae* (Table 1) [16,17]. Disrupting guanine riboswitch regulated gene expression leads to a compromised growth rate of *B. subtilis* [96]. Further, attempts to generate a *B. subtilis* strain that lacks all transcriptional units under the regulation of the guanine riboswitch failed. Altogether, these experiments suggest that the guanine riboswitch is a suitable target for narrow-spectrum antibiotics.

The guanine riboswitch binds to its cognate ligand with high affinity (~4 nM) and discriminates strongly against adenine (~10,000-fold). The three-dimensional structures of the riboswitch in complex with various ligands showed almost complete encapsulation of ligands accompanied by local conformational changes around the binding pocket and extensive interactions through multiple hydrogen bonds with U22, U47, U51 and C74 (Figure 5a) [37,97,98]. The high specificity to guanine is achieved by Watson–Crick base pairing with the nucleobase C74. Based on these observations, efforts to discover the guanine riboswitch targeting compounds have remained focused on purine analogues.

Breaker and colleagues were the first to examine the antibacterial activity of guanine analogues [96]. Although the study identified 2-amino-N6-hydroxyadenine (Figure 5b) to exert antibacterial activity by targeting the guanine riboswitch, the possibility of this known mutagen hitting off-target(s) could not be excluded [99,100]. Lafontaine et al. explored the guanine riboswitch by screening various pyrimidine derivatives and guanine analogues modified at the two-, and six-positions or in the five-membered ring [101,102]. The most promising hit was 2,5,6-triaminopyrimidin-4-one (PC1, Figure 5b). Interestingly, PC1 inhibited the growth of only those clinical strains wherein guanosine monophosphate (GMP) synthetase (*guaA*) is under riboswitch control, namely *S. aureus*, *S. haemolyticus*, methicilin resistant *S. aureus COL* and *C. difficile*. Further, in mice, as well as a bovine model of *S. aureus* infection, PC1 reduced the bacterial burden in a dose-dependent manner [101,103]. However, it is unclear if the antibacterial effect is only due to binding to the guanine riboswitch. One concern is that no resistant bacteria were obtained even after > 30 passages of a *S. aureus* strain containing functional *guaA* regulated by a riboswitch [101], which can mean that PC1 has multiple targets. Further, in a different study it was found that guanine and GMP only marginally rescued bacterial growth in the presence of PC1, implying that the guanine pathway is not the sole target of this compound [104]. It should, however, be noted that different *S. aureus* strains were used in these studies so that strain-dependent variations in response to PC1 cannot be excluded. However, PC1 was also shown to be cytotoxic to macrophages, which lack a guanine riboswitch further suggesting that PC1 has other targets besides the guanine riboswitch. Regardless, the discovery of PC1 advocates the possibility of targeting the guanine riboswitch with compounds containing a distinct chemical scaffold from the cognate ligand. As the guanine riboswitch regulates genes involved in *de novo* guanine biosynthesis—*guaA* and *guaB*—which are only present in a few clinical pathogenic strains, the riboswitch is considered a target for narrow-spectrum antibacterials.

## 6. General Considerations for Riboswitch Drug Discovery

Selectively targeting RNA with small molecules for therapeutic purposes can be an insurmountable challenge. Targeting the well-formed pockets of riboswitches evolved to bind to small molecules offer an immense opportunity to achieve this goal. Indeed, the examples discussed above have established riboswitches as novel antibacterial targets. Using diverse strategies such as fragment screening, HTS, phenotypic screening and rational design of structural analogues, molecules with antibacterial effects have been identified for the FMN, TPP, and guanine riboswitch.

Despite the promise riboswitches present, various challenges restrict riboswitch-targeting drug discovery. While some of these challenges are associated with the riboswitch itself, others could be overcome by developing riboswitch-specific methods for hit screening and lead development, and characterization of their mode of action.

For a riboswitch class to be considered a potential antibacterial target, mere occurrence or prevalence is not sufficient. Instead, the essentiality of the regulated gene(s) is the obligatory condition. For example, while antibiotic compounds targeting the *lysC* riboswitch regulating lysine biosynthesis were discovered [105], the riboswitch is not considered a viable target because many bacteria harbour isozymes that are not regulated by the riboswitch [106]. Further, many pathogens can sequester lysine from the host, rendering inhibition of lysine biosynthesis ineffective [107,108]. Lafontaine and colleagues demonstrated that their most promising guanine riboswitch ligand PC1 (Figure 5b) only acts against clinically relevant bacteria that have *guaA* under riboswitch regulation [101]. Similarly, the Roemer group showed that the antibacterial compound ribocil inhibits only those bacterial species wherein the FMN riboswitch regulates the *ribB* gene involved in riboflavin *de novo* biosynthesis [61]. The riboswitch abundance which varies from one bacterial species to another species could also influence the potency of the potential antibiotic in a strain-dependent manner. Strains containing the same riboswitch aptamer at multiple and crucial genetic loci could be more sensitive than strains with a riboswitch at only a single, or fewer loci. Unfortunately, no studies have yet been conducted to address the effect of riboswitch abundance on potency of potential antibiotics. Collectively, prior knowledge of the genes regulated by a riboswitch in various pathogens and the abundance of the riboswitch is crucial for its success as an antibacterial target. Finally, some riboswitches are kinetically controlled, while others are thermodynamically controlled [109,110,111,112]. For drug discovery, it would be helpful to understand how targeting riboswitches with the different control mechanism affects the MIC against bacterial pathogens.

Another important consideration for riboswitches as drug targets is resistance development. Bacteria are known to acquire or develop antibiotic resistance through various mechanisms that fall into three groups: (1) by minimizing their intracellular steady-state concentration to prevent access to the target, (2) by modifying the molecular target through genetic mutation or post-translational mechanism and (3) by inactivating the antibiotic through hydrolysis or modification [113,114]. For compounds targeting riboswitches, only the second of these mechanisms has been observed [45,61,62,64]. Riboswitch abundance could be an important factor in this context [17,115]. In general, it can be conceived that bacteria with abundant riboswitches regulating more than one essential gene cluster should have a relatively low FOR against a potential antibiotic, as this would require bacteria to gain mutations at multiple genomic loci. However, we are not aware of any studies in which this aspect has been investigated. Intriguingly, the riboswitch sequence also appears to affect FOR. For example, the FOR to ribocil against *E. coli* (2.4 × 10^−6^) is higher than that of heterologous strains containing the FMN riboswitch from *P. aeruginosa* (6.4 × 10^−7^) and *A. baumannii* (3.3 ×10^−8^) instead of the native chromosomal copy. The reasons for this are unclear. Clearly, more studies are needed to assess resistance development against riboswitch ligands in vivo.

Most of the lead molecules identified for riboswitch targets are structural analogues of their cognate ligands. As the cognate ligands are often the end product of the metabolic pathway, these potent structural analogues could potentially also be recognized by the intracellular enzymatic machinery leading to undesired intracellular off-targets in the human host [116]. The serendipitous discovery of ribocil through phenotypic screening suggests that riboswitches can also selectively bind ligands with chemical scaffolds distinct from the natural ligand. Computational studies have suggested that some riboswitch pockets have similar properties in terms of size and polarity than druggable protein binding sites [42,43,44]. The TPP, FMN and SAM-I riboswitches only partially enclose their rather large ligands to achieve binding affinities in the low nanomolar range and thereby allow more flexibility in terms of exploring the chemical space. On the contrary, the purine and *lysC* riboswitches almost entirely encapsulates their ligand heavily restricting the possibility of identifying ligands with chemical scaffold distinct from their cognate ligands. It is highly likely that with a better exploration of chemical space than currently completed for some riboswitches, potent ligands that differ considerably from the cognate metabolite could be discovered, as already exemplified by ribocil. However, in order to investigate this possibility, screening of a large number of fragments and/or compounds against each of these promising riboswitches is crucial which in turn demands to develop and to implement riboswitch specific modern drug discovery approaches. In the following, we review various methods developed for riboswitch-oriented high-throughput screening (HTS), fragment-based screening and structure-based virtual screening.

### 6.1. High-Throughput Screening (HTS)

Although the development of HTS methods for riboswitches has lagged behind those for proteins, gradual advancement has been reported. Some of the developed assays monitor riboswitch activity, either in vitro or *in vivo*. In the first such effort, Mayer et al. developed a fluorescence polarization (FP)-dependent screening assay for the *glmS* riboswitch and demonstrated its robustness screening ~ 5000 commercial compounds in a 96-well plate format [92]. The riboswitch was labelled at the 5′ end with a fluorophore (fluorescein) to measure ligand-induced *in-cis* cleavage. Around the same time, the Breaker group developed a fluorescence resonance energy transfer (FRET) assay for the same riboswitch. They used two fluorophores tagged at either the 5′ or 3′ -end of the *glmS* riboswitch [90]. In the off-state, the fluorescence from the fluorophore at the 5′ end is quenched by the fluorophore at the 3′ end. However, the ligand-induced cleavage separates the two fluorophores leading to an increase in the fluorescence signal from the 5′ fluorophore. The suitability of the assay for HTS was demonstrated by screening 960 compounds in a 384-well plate format. Although these studies established that HTS could successfully be applied to the *glmS* riboswitch, the assay principles could not be transferred to other riboswitch classes as these do not act as ribozymes. An alternative HTS-compatible assay for riboswitches that regulate translation was developed by Lünse et al. They developed a TPP riboswitch translational fusion construct with β-galactosidase. An *E. coli* strain was transformed with the construct and exponentially grown cultures of transformants were incubated with compounds. This was followed by cell lysis to report β-galactosidase expression by colorimetric estimation of O-nitrophenyl-β-galactoside (ONPG) breakdown [117]. Another assay that screens for riboswitch ligands in a biological context was developed by Schneider et al. They engineered a *B. subtilis* strain to have: 1) a transcriptional fusion of the guanine riboswitch and the transcriptional repressor *blaI* gene under the control of a xylose inducible promoter and 2) the *luxABCDE* genes under the control of a BlaI responsive promoter (P_blaP_) [118]. Binding of ligands to the guanine riboswitch causes transcriptional termination and therefore no *blaI* is expressed. This, in turn, allows transcription of *luxABCDE* genes through P_blaP_ leading to an increase in the luminescence signal. The assay allows to monitor riboswitch activity in vivo without the need for cell lysis. The suitability of the assay for HTS was demonstrated by screening a 1200-compound library in 384-well plate format. Further, using the adenine (*pbuE*) riboswitch as a model, Chinnappan et al. demonstrated that dual molecular beacons can be used to monitor a riboswitch regulated transcription reaction [119]. The assay relies on two oligonucleotides, each of them labelled with a unique fluorophore and a quencher (also called beacons). One of these oligonucleotides is complementary to a sequence upstream of the aptamer domain sequence (called 5′-beacon) and another one to a sequence downstream to the terminator stem of the riboswitch (called 3′-beacon). When the riboswitch is being transcribed, the 5′-beacon binds to the target sequence and thus unquenches the fluorophore. The second beacon can only bind (and thus results in an increase in fluorescence), if the riboswitch regulated transcription is switched on. Thus, the assay specifically screens for ligands that modulate the transcription activity of a riboswitch. While the principle function of the assay has been demonstrated, it has not yet been used for screening a larger library. In summary, different approaches to monitor riboswitch activity have been developed and some of the assays have been used to screen compound libraries. However, no attractive hits have yet been discovered. This could be due to the fact that the screening libraries were rather small and thus lacked diversity.

Another strategy for riboswitch hit discovery is to screen for binding to the aptamer domain only. Rizvi et al. employed affinity selection mass spectrometry (AS-MS) for detection of selective small molecule-riboswitch binding. Using the FMN riboswitch as a model system, they demonstrated the feasibility of this approach for HTS by screening a library of ~53,000 compounds leading to the identification of ~20 specific hits (< 0.04% hit rate) [76]. More recently, the Schneekloth group employed a small molecule microarray (SMM) technology to screen a library of ~26,000 compounds for preQ1 riboswitch binding and identified one promising hit (K_D_ = 0.5 μM) [120]. Subsequently they determined the binding mode of the compound and initiated a small SAR study to improve the potency of the hit.

### 6.2. Fragment-Based Screening

There are only limited examples of riboswitch-oriented fragment-based screening, probably because screening RNA targets is generally considered to be challenging [121] Cressina et al. used a hierarchical approach to discover TPP riboswitch binding fragments. First, they employed equilibrium dialysis to screen a library of 1300 fragments against the TPP riboswitch resulting in 20 hits [55]. Subsequently, binding was confirmed using NMR analysis, isothermal titration calorimetry and counter screening against the unrelated lysine riboswitch. Thus, 10 compounds, which were selective for the TPP riboswitch, were discovered. For four of them, the binding modes were subsequently determined using X-ray crystallography (Figure 2c). Recently, Binas et al. demonstrated the application of ^19^F NMR for hit identification using a library of 102 fragments against 14 RNA targets of different sizes and architectures including riboswitches [122].

### 6.3. Structure-Based Virtual Screening

Finally, structure-based virtual screening (SBVS) offers a powerful means to screen for potential ligands from a chemical database of millions of molecules. While various SBVS algorithms, originally developed for protein targets, can be applied to RNA targets, their application requires RNA specific re-parameterization of scoring function or the development of RNA-specific approaches. Towards this end, our group and others have made some advancements and discovered novel ligands against purine, SAM and lysine riboswitches [123,124,125,126]. The developments in RNA-targeted virtual screening are discussed in detail elsewhere [121,127].

In conclusion, various approaches for screening ligands using functional assays have been developed, but none of the conducted screens has delivered any hits. One major reason for this is probably that the investigated libraries were rather small and lacked diversity. Using larger libraries to explore the chemical space of riboswitch ligands appears to be justified. In contrast, screening just the riboswitch aptamer domain for binders delivered hits using various approaches. Notably, the hit compounds possess scaffolds which are chemically distinct from the natural ligands. However, especially when fragment libraries where screened, the hits had a rather low affinity and optimization to increase affinity is necessary. In the case of a hit discovered for the preQ1 riboswitch using SMM technology successful affinity optimization was achieved with just a few compounds, suggesting that such a strategy is very promising.

## 7. Conclusions

Riboswitches constitute attractive RNA targets for next generation antibiotics with a novel mechanism of action. Undoubtedly, the FMN riboswitch is the one that has been most explored as an antibacterial target. The discovery of ribocil C is probably the most important contribution to the field, as it established the possibility of targeting a riboswitch with synthetic ligands having chemically distinct scaffold as compared to their natural ligands. Identification and optimization of such synthetic ligands could significantly reduce the risk of hitting off-target(s). Another interesting contribution to exploring riboswitches as antibacterial drug targets is the invention of 5FDQD, which demonstrated that negatively charged groups could be replaced with hydrophobic moieties without compromising the affinity. However, the high frequency of resistance against these ligands remains a serious concern. The widespread occurrence of the TPP riboswitch in pathogenic bacteria, together with its binding site that appears to be well suited for drug-like compounds, makes this riboswitch a very promising target. Thus far, the investigated changes of the natural ligand have severely compromised affinity, however the diverse fragment hits could serve as starting points to explore this riboswitch further.

In general, to thoroughly investigate the promise of riboswitches as antibacterial drug-targets, focused and integrated efforts from chemical biologists and microbiologists are crucial. Despite their relatively recent discovery, significant developments have been made in this direction. The studies discussed here not only point to the potential of riboswitches as antibacterial targets but also reveal various challenges that need to be addressed to further advance the field. In particular, there is a high need to thoroughly explore the chemical space of riboswitch ligand in order to identify selective compounds with chemical scaffolds distinct from the cognate natural ligands. To achieve this goal, the development of robust high-throughput screening assays and the screening of large compound collections will be crucial.

## Figures and Tables

**Figure 1 antibiotics-10-00045-f001:**
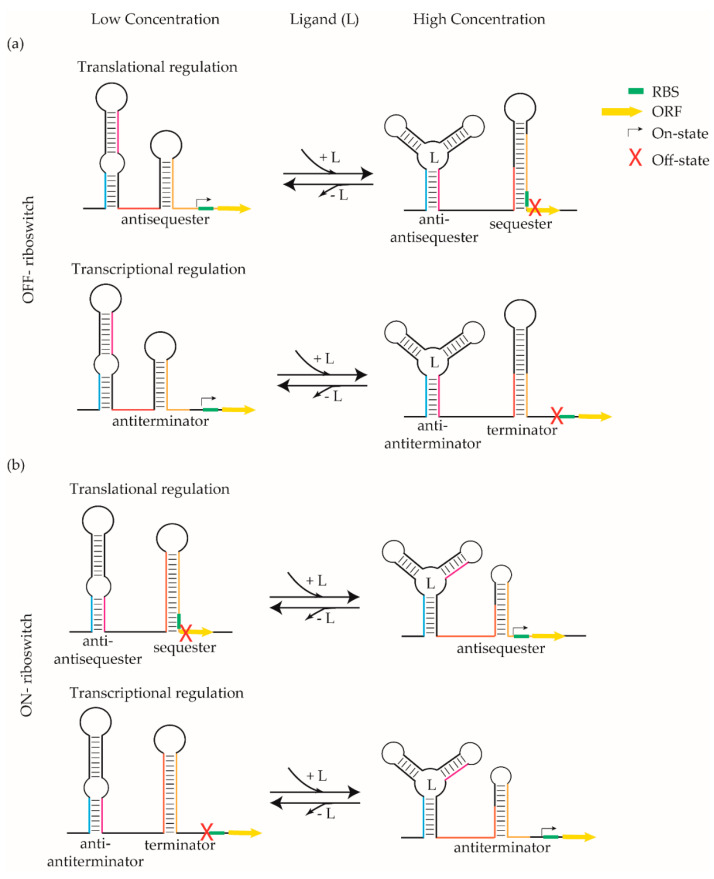
Riboswitch-mediated gene regulation. Bacterial riboswitches regulate the expression of immediate downstream genes by modulating their conformation in response to a change in the concentration of their cognate ligands (L). (Strands involved in the formation of anti-antiterminator/anti-antisequester (cyan and magenta) and terminator/sequester (orange and yellow) are color-coded). In most cases, binding of ligand leads to transcriptional and/or translational termination of downstream gene expression (open reading frame (ORF), marked in yellow). These riboswitches are known as “OFF-switches” (**a**). In other cases, ligand binding leads to their expression of downstream genes and these riboswitches are called “ON-switches” (**b**). In the case of translation regulation, the RBS (green box) is either sequestered (**a**) or released (**b**) upon ligand binding. In the case of transcriptional regulation, the RNA polymerase is either stalled by a terminator (**a**) or moves past the antiterminator (**b**) upon ligand binding. RBS: ribosomal binding site.

**Figure 2 antibiotics-10-00045-f002:**
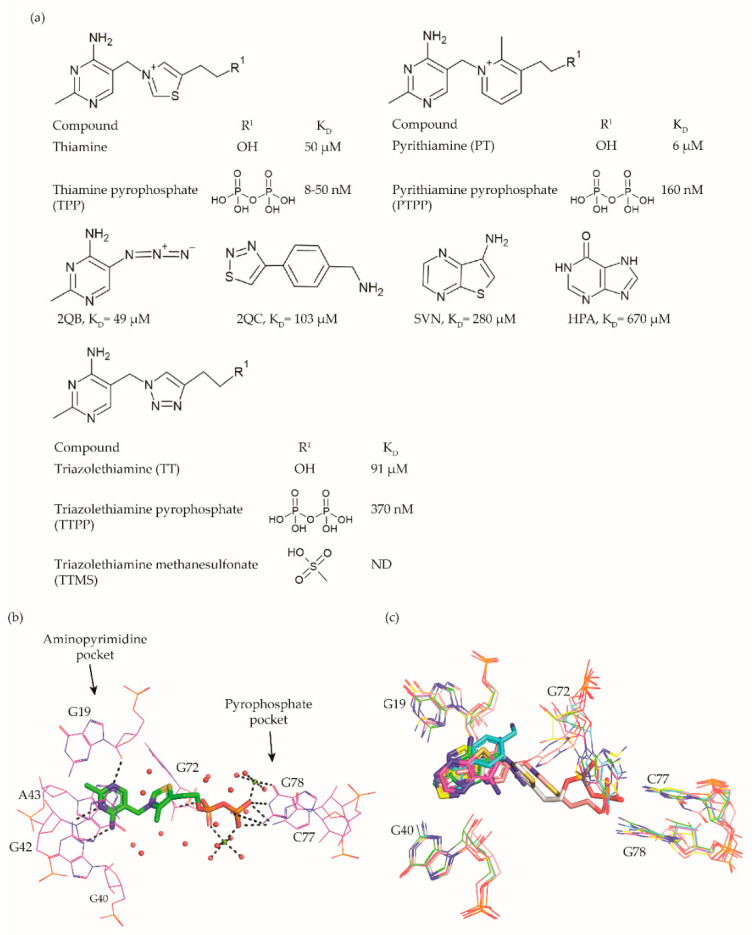
TPP riboswitch ligands. (**a**) Chemical structures and binding affinities of ligands taken from various studies [32,45,52,53]. (**b**) The binding pocket of the TPP riboswitch in complex with TPP (Protein Databank (PDB) ID: 2GDI). The aminopyrimidine and pyrophosphate binding pockets are labelled. Water molecules are shown as red spheres, magnesium ions as green spheres, and hydrogen bonds as dotted lines. The negative charge on TPP is compensated by magnesium ions through direct and indirect hydrogen bonds with the surrounding residues. (**c**) Superposition of TPP riboswitch binding pocket in complex with different fragments (PDB ID: 4NYA (green), 4NYB (cyan), 4NYC (magenta), 4NYD (yellow), 4NYG (pink) and 2GDI (grey)). The conformational flexibility of G72 is visible.

**Figure 3 antibiotics-10-00045-f003:**
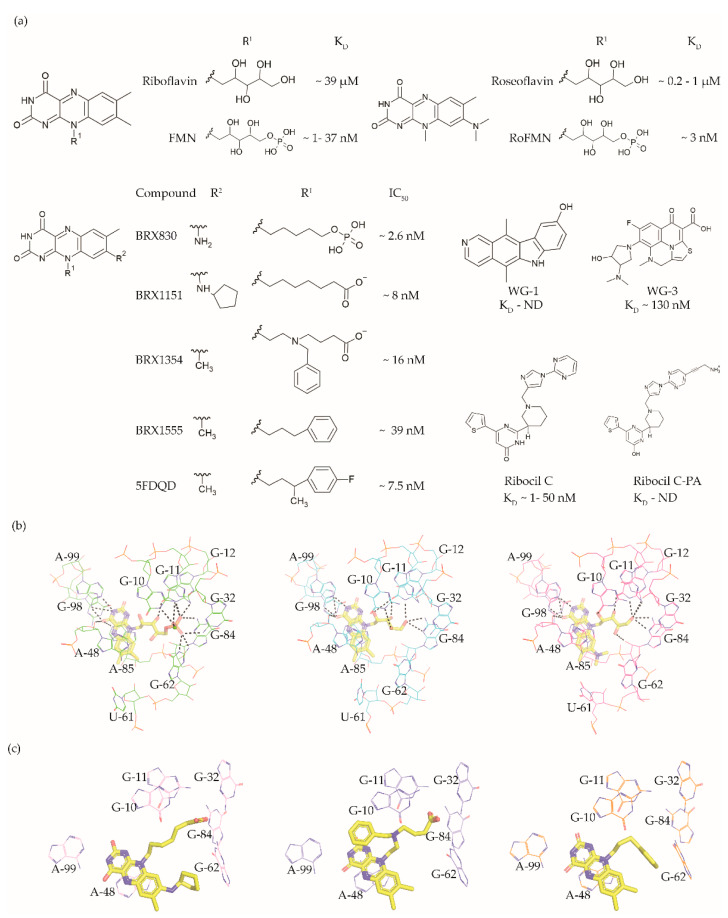
FMN riboswitch ligands. (**a**) Chemical structures of the FMN riboswitch ligands together with K_D_ or IC_50_ values taken from various studies [34,61,62,63,64]. (**b**) The binding pocket of the FMN riboswitch in complex with FMN (PDB ID: 3F2Q, left), riboflavin (PDB ID: 3F4G, middle) and roseoflavin (PDB ID: 3F4H, right). The polar contacts are shown as black dotted lines and the magnesium ion as a green sphere. (**c**) The binding poses of various ligands within the FMN riboswitch binding pocket—BRX1151 (PDB: 6DN1, pink), BRX1354 (PDB: 6DN2, blue), and BRX1555 (PDB: 6DN3, orange).

**Figure 4 antibiotics-10-00045-f004:**
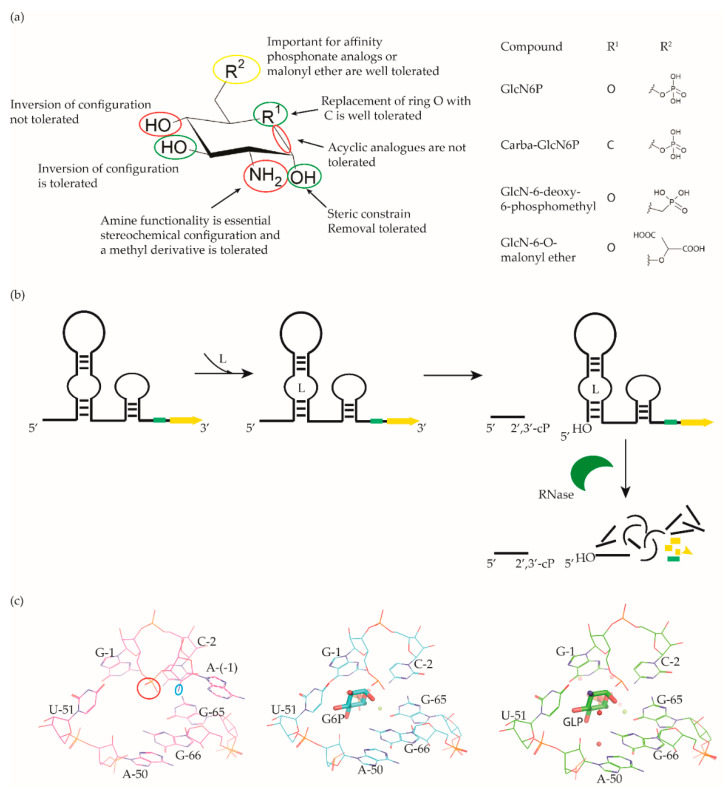
*glmS* riboswitch ligands. (**a**) Chemical structures and structure–activity relationships (SAR) of the GlcN6P derivatives. Positions where changes are well tolerated are marked in green, moderately tolerated in yellow and not tolerated in red. (**b**) Cartoon representation of ligand-dependent *glmS* ribozyme activity. The RBS is shown as a green bar and the ORF in yellow. Binding of ligand induces *glmS* riboswitch self-cleavage to produce an upstream fragment with a 2′,3′- cyclic phosphate end (2′,3′-cP) and a downstream fragment with a 5′-OH end, followed by degradation of the latter by a specific RNase. (**c**) Snapshot of the binding pocket of the *glmS* riboswitch in the apo form (PDB ID: 2GCS, left), in complex with glucose-6-phosphate (PDB ID: 2Z74, center), and GlcN6P (PDB ID: 2Z75, right). The nucleophile that ends up in the upstream fragment (2′-OH group of A(-1)) is highlighted with a blue circle and the leaving group (5′-phosphate group of G1) with a red circle.

**Figure 5 antibiotics-10-00045-f005:**
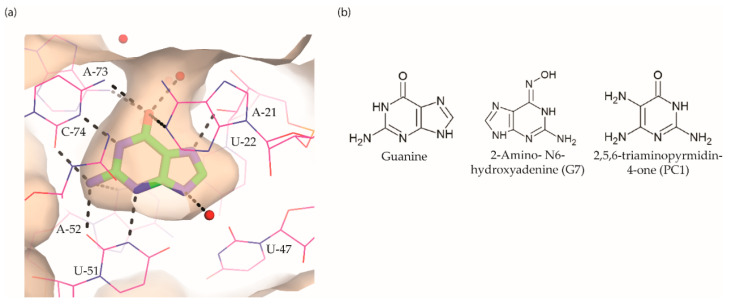
Guanine riboswitch ligands. (**a**) The binding pocket of the guanine riboswitch in complex with guanine (PDB ID: 6UBU). The solvent accessible surface is shown in wheat colour, the ligand with green colour, polar contacts as black dotted lines and water molecules as red spheres. (**b**) Chemical structure of the guanine riboswitch ligands.

**Table 1 antibiotics-10-00045-t001:** Prevalence of riboswitches in WHO priority list pathogens. The prevalence of each riboswitch among the WHO priority list pathogens was obtained from the Rfam database [24] using the riboswitch class and bacterial species as query key words and is listed in brackets. The number in brackets represents the range of riboswitch abundance in different strains of bacterial species.

Riboswitch	Type	Cognate Ligand	Prevalence
FMN	Off	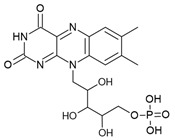 flavin mononucleotide (FMN)	*Acinetobacter baumannii* (1)*, Pseudomonas aeruginosa* (1), *Enterobacteriaceae*, *Enterococcus faecium* (1–2), *Staphylococcus aureus* (2), *Streptococcus pneumoniae* (1–2), *Haemophilus influenzae* (1), *Shigella* spp. (1)
*c*-di-AMP	Off	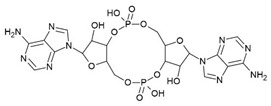 cyclic diadenine monophosphate (*c*-di-AMP)	*Mycobacterium tuberculosis (1)*
Fluoride	On	F^-^	*A. baumannii* (2)*,**P. aeruginosa* (2),*E. faecium* (1–3),
*glmS*	Off	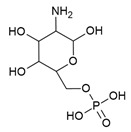 glucosamine-6-phosphate (GlcN6P)	*S. aureus* (1), *E. faecium* (1)
Glycine	On	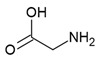 glycine	*H. influenzae* (1),*S. pneumoniae* (1),*Neisseria gonorrhoeae* (1),***S. aureus* (1), *A. baumannii* (1)*, M. tuberculosis* (2)
Lysine	Off	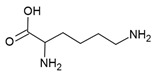 lysine	*E. faecium* (1), *S. aureus* (2), *H. influenzae* (1), *Shigella* spp. (1)
Moco	Off	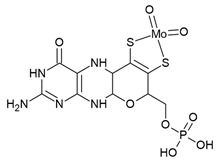 molybdenum cofactor (Moco)	*Enterobacteriaceae*,*H. influenzae* (2),*Shigella* spp. (2)
PreQ1	Off	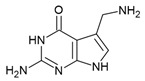 7-aminomethyl-7-deazaguanine (preQ_1_)	*E. faecium* (1–2),*N. gonorrhoeae* (1),*S. pneumoniae* (1),*H. influenzae* (1)
Purine	Off	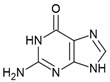 guanine	*E. faecium* (1),*S. aureus* (1)
On	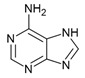 adenine
SAM	Off	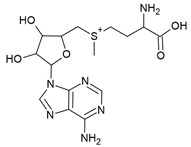 S-adenosylmethionine (SAM)	*S. aureus* (1–4)*, N. gonorrhoeae* (1), *M. tuberculosis* (1)
TPP	Off	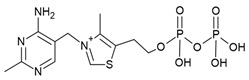 thiamine pyrophosphate (TPP)	*A. baumannii* (1), *P. aeruginosa* (1), *Enterobacteriaceae, E. faecium* (2), *S. aureus* (1–2)*, H. pylori* (1), *Campylobacter* spp. (1–2), *Salmonellae* (1–3), *N. gonorrhoeae* (2), *S. pneumoniae* (1–5)*, H. influenzae* (3), *M. tuberculosis* (2) and *Shigella* (3)

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
