# Peer review of "Riboswitches as Drug Targets for Antibiotics"

_antibiotics, 2021, doi:10.3390/antibiotics10010045_

Round 1

Reviewer 1 Report

This paper reviewed riboswitches as drug targets for antibiotics by introducing the TPP, FMN, glmS, guanine and other riboswitch ligands as antibacterial targets.

There are some minor concerns:

  1. It will be good to summarize the difference, advantage, and disadvantage of these riboswitch ligands on antibiotic discovery.
  2. Line 49 “… gene expression (open reading frame (ORF), marked in green).” But in Figure 1. ORF is marked in yellow.
  3. Line 272 What is “MIC”?
  4. Line 281 What is “WG-1”?
  5. Line 307 “RNAse” is not the same as “Rnase” in Figure 4.
  6. Line 355 “4. Guanine riboswitch ligands” should be “5. Guanine riboswitch ligands”?
  7. Line 382 What is “PC1”?

Reviewer 2 Report

Comment 1: There are several mechanism by which bacteria develops antibiotic resistance. These mechanisms need to be discussed as well. 

Comment 2: Consequently, a need for a new generation ..... recognized Describe in details. What are the new generation of antibiotics??

Comment 3: "Riboswitches as drug targets for antibiotics" Why only antibiotics, it may be any other small molecule as well.

Comment 4: A discussion on the use of modern drug discovery approaches toward the design of riboswitch drug will be a valuable addition.

Comment 5:Table 1 needs revision as content from row is missing.

Comment 6: Most of the abbreviations are not explained when they are mentioned for the first time.

Round 2

Reviewer 2 Report

The authors have addressed all the points.